# Looking at Marine-Derived Bioactive Molecules as Upcoming Anti-Diabetic Agents: A Special Emphasis on PTP1B Inhibitors

**DOI:** 10.3390/molecules23123334

**Published:** 2018-12-15

**Authors:** Shahira M. Ezzat, Mahitab H. El Bishbishy, Solomon Habtemariam, Bahare Salehi, Mehdi Sharifi-Rad, Natália Martins, Javad Sharifi-Rad

**Affiliations:** 1Pharmacognosy Department, Faculty of Pharmacy, Cairo University, Kasr El-Ainy Street, Cairo 11562, Egypt; shahira.ezzat@pharma.cu.edu.eg; 2Department of Pharmacognosy, Faculty of Pharmacy, October University for Modern Science and Arts (MSA), Cairo 12566, Egypt; mahelmy@msa.eun.eg; 3Herbal Analysis Services UK & Pharmacognosy Research Laboratories, University of Greenwich, Central Avenue, Chatham-Maritime, Kent ME4 4TB, UK; s.habtemariam@gre.ac.uk; 4Student Research Committee, Bam University of Medical Sciences, Bam 44340847, Iran; 5Department of Medical Parasitology, Zabol University of Medical Sciences, Zabol 61663-335, Iran; 6Institute for Research and Innovation in Health (i3S), University of Porto, 4200-135 Porto, Portugal; 7Faculty of Medicine, University of Porto, Alameda Prof. Hernâni Monteiro, 4200-319 Porto, Portugal; 8Zabol Medicinal Plants Research Center, Zabol University of Medical Sciences, Zabol 61615-585, Iran; 9Department of Chemistry, Richardson College for the Environmental Science Complex, The University of Winnipeg, 599 Portage Avenue, Winnipeg, MB R3B 2G3, Canada

**Keywords:** protein-tyrosine phosphatase 1B, type 2 diabetes mellitus, insulin signaling pathways, marine metabolites

## Abstract

Diabetes mellitus (DM) is a chronic metabolic disease with high morbimortality rates. DM has two types: type 1, which is often associated with a total destruction of pancreatic beta cells, and non-insulin-dependent or type 2 diabetes mellitus (T2DM), more closely associated with obesity and old age. The main causes of T2DM are insulin resistance and/or inadequate insulin secretion. Protein-tyrosine phosphatase 1B (PTP1B) negatively regulates insulin signaling pathways and plays an important role in T2DM, as its overexpression may induce insulin resistance. Thus, since PTP1B may be a therapeutic target for both T2DM and obesity, the search for novel and promising natural inhibitors has gained much attention. Hence, several marine organisms, including macro and microalgae, sponges, marine invertebrates, sea urchins, seaweeds, soft corals, lichens, and sea grasses, have been recently evaluated as potential drug sources. This review provides an overview of the role of PTP1B in T2DM insulin signaling and treatment, and highlights the recent findings of several compounds and extracts derived from marine organisms and their relevance as upcoming PTP1B inhibitors. In this systematic literature review, more than 60 marine-derived metabolites exhibiting PTP1B inhibitory activity are listed. Their chemical classes, structural features, relative PTP1B inhibitory potency (assessed by IC_50_ values), and structure–activity relationships (SARs) that could be drawn from the available data are discussed. The upcoming challenge in the field of marine research—metabolomics—is also addressed.

## 1. Introduction

Diabetes mellitus (DM) is considered a major health problem worldwide [1]. Obesity and diabetes incidence still continue increasing due to globalization, mechanization, and changes in human lifestyle and daily routines [2]. According to the International Diabetes Federation (IDF), it was estimated that in 2017 there were 451 million (age 18–99 years) people with diabetes worldwide. These figures were expected to increase to 693 million by 2045 [3]. DM is a chronic metabolic disease that results from defects in insulin action, insulin secretion, or both, leading to persistent hyperglycemia [4]. 

Currently, type 2 diabetes mellitus (T2DM) represents a major threat to health [5]. Characterized by increased blood glucose levels, this is the underlying reason for several complications, including cardiovascular disorders, blindness, kidney failure, and peripheral nerve damages [6]. The development of T2DM and its complications are related, in most cases, to insulin resistance and postprandial hyperglycemic variations [7,8]. Thus, an effective drug for controlling insulin resistance may be beneficial in improving the quality life of T2DM patients. Several pharmacological strategies have been investigated on DM treatment, including insulin release stimulation, gluconeogenesis inhibition, glucose transport activity increase, and intestinal glucose absorption reduction [9]. Insulin supplements and other oral anti-diabetic drugs can be used alone or in combination to improve glycemic regulation [10]. However, some of the available anti-diabetic drugs have either the disadvantage of having low efficacy or serious side effects [11]. Thus, there is a continuous search for more effective and safer anti-hyperglycemic agents, especially from natural origins.

Insulin sensitizers, such as thiazolidinediones (TZDs or glitazones) have been used as effective drugs for T2DM treatment [12]. The identification of the enzyme responsible for the dephosphorylation of insulin receptors, called protein-tyrosine phosphatase 1B (PTP1B), showed that the inhibitors of such an enzyme could be employed as insulin sensitizer agents and, therefore, as promising anti-diabetic drugs [13]. This hypothesis was confirmed in mouse models, where it was found that PTP1B gene disruption can increase insulin sensitivity. Similar results were also obtained when PTP1B antisense nucleotides suppressed PTP1B gene expression [14].

Protein tyrosine phosphatases (PTPs) constitute a huge and structurally variable family of highly regulated enzymes. Most PTPs have been proposed to be targets for advanced drug discovery, and PTP1B is one of the well-established enzymes among PTPs [15,16,17]. It was the first isolated member of the PTP superfamily, and since then, growing evidence has linked it with insulin resistance, obesity, and T2DM. Numerous studies have shown that PTP1B can negatively regulate insulin and leptin signaling pathways. Indeed, PTP1B dephosphorylates both insulin receptor and its substrate IRS-1 in the insulin signaling pathway [18,19], whereas in the leptin pathway, PTP1B binds and dephosphorylates tyrosine kinase downstream of the Janus-Activated Kinase 2 (JAK2) leptin receptor [20]. In cell cultures, PTP1B overexpression causes a decrease in the insulin-stimulated phosphorylation of IR and IRS-1, while PTP1B raises insulin-initiated signaling level reduction [21,22]. The hypothesis that PTP1B expression can contribute to diabetes and obesity is supported by quantitative analysis of trait loci and mutations in the human PTP1B gene [23]. In in vivo studies, PTP1B knockout mice exhibited elevated resistance to high-fat diet-induced obesity and insulin sensitivity [24,25]. In addition, other studies on tissue-specific PTP1B knockout mice have shown that leptin action, adiposity, as well as body weight are controlled by neuronal PTP1B [26]. Generally, many studies suggest that PTP1B inhibitors constitute a highly promising approach for T2DM and obesity amelioration.

Aryl carboxylic acids, such as isoxazole [27], hydroxylpropionic [28], 2-oxalylamino benzoic (OBA) acids [29], and thiophene diacid [30], have been recognized as alternative phosphotyrosine (pTyr) surrogates to overcome the lack of cellular activity of highly charged phosphonates (Figure 1). Furthermore, it was reported that benzyl aryl α-ketoacid derivatives revealed significant PTP1B inhibitory effects in a non-competitive pattern, targeting conserved protein loop (WPD loop) open conformation [31]. It was also noticed that the existence of a benzyl group in these bioactive molecules may enhance PTP1B binding affinity, and being hydrophobic in nature, it also increases their cell membrane permeability. Recent studies also suggested that PTP1B may become an oncogene in breast cancer [16]. Accordingly, multiple studies have been conducted focusing on the development of new PTP1B inhibitors for the treatment of T2DM, obesity, and cancer, but to the authors’ knowledge there are no review articles published on this subject. In this sense, the present review aims to provide an overview of the role of PTP1B in T2DM insulin signaling and treatment, and to highlight the most recent findings on several compounds and extracts discovery from marine organisms and their relevance as upcoming PTP1B inhibitors.

## 2. Marine Sources as Upcoming Therapeutic Agents

The marine environment is considered a wide and relatively unexploited source of bioactive compounds with high biodiversity, including fatty acids (especially polyunsaturated fatty acids), proteins, polyphenols, sterols, sulfated polysaccharides, and pigments [32,33,34,35,36]. Indeed, marine algae has been increasingly exploited as renowned sources of metabolites with promising biological effects, including antioxidant, hypoglicemic, hypotensive, hypolipidemic, antibacterial, and antiviral activities [37,38]. Specifically, macroalgae are considered healthy foods as they are rich in minerals and dietary fibers. Traditionally, the Far East and Hawaiian Islands, Japan, Korea, and China consume marine algae as a common component of their diets. Macroalgae species can reach 9000 species and can be classified according to their pigment composition into three classes, i.e., *Phaeophyta, Rhodophyta*, and *Chlorophyta* (also known as brown, red, and green algae, respectively) [39].

Unique metabolites from diverse classes have been isolated from different marine plants, with in vivo remarkable pharmacological effects [40], such as anticancer, anti-hyperlipidemic, anti-diabetic, anti-hypertensive, antioxidant, anti-inflammatory, anticoagulant, anti-estrogenic, antibacterial, antifungal, antiviral, immunomodulatory, neuroprotective, and tissue healing properties [41]. More recently, as a result of the characterization of a large number of bioactive metabolites from marine macroalgae, there has been a growing interest in the search for potential applications of macroalgae and their metabolites as functional constituents for human and animal health benefits [42]. Functional constituents of macroalgae have been increasingly used as food supplements as well as for anti-diabetic purposes [40]. Hereby, the possible applications of marine macroalgae and/or macroalgae-derived bioactive metabolites for PTP1B inhibitory effects have been greatly expanded.

## 3. Marine-Derived Molecules with PTP1B Inhibitory Activity

### 3.1. Ptp1b Inhibitory Activity: In Vitro Findings

Around 300 natural products with PTP1B inhibitory capacity were isolated and characterized from different natural sources, many of them from marine origin [43]. The isolation and identification of sulfircin, a sesterterpene sulfate, from deep-water sponge *Ircinia* (unknown species), was the first documented marine metabolite possessing PTP1B inhibitory activity [43]. Since then, marine sponges have been considered valuable sources of PTP1B inhibitors with diverse structures [44], such as polybromodiphenyl ether [45], sesquiterpenoids, and sesquiterpene quinones [46]. Nevertheless, the novelty of marine resource screening models has encouraged the development of new studies targeting these resources as upcoming anti-diabetic agents. Marine algae, seaweeds, soft corals, sponges and lichens are considered to be among these models as they were found to exhibit PTP1B inhibitory effects. Table 1, Table 2, Table 3, Table 4, Table 5, Table 6, Table 7 and Table 8 summarize a large number of isolated compounds from marines that have PTP1B inhibitory effects with varying potencies. In the following sections, the PTP1B inhibitory activity of some of these compounds are discussed.

#### 3.1.1. Bromophenols

As the main component of algae, bromophenols may be responsible for the reported anti-diabetic activity of many marine organisms. These compounds arise from the tendency of the phenol moiety to undergo electrophilic bromination to varying degrees (Table 1).

Bromophenols (compounds **1**–**11**) isolated from the red algae *Rhodomela confervoides* have potent in vitro PTP1B inhibitory effects, with IC_50_ values fluctuating between 0.8 μM and 4.5 μM [47,48,49,50,51,52,53,54]. This change in potencies could be attributed to the bromine content of these compounds or to their side chains. On the other hand, Yamazaki et al. [45] isolated two bromophenols (compounds **12** and **13**) from the Indonesian marine sponge *Lamellodysidea herbacea* and found positive in vitro PTP1B inhibitory effects, with IC_50_ values of 0.9 μM and 1.7 μM, respectively. Other brominated phenols (compounds **14**–**16**) isolated from red algae *Symphyocladia latiuscula* by Liu et al. in 2011 [55] also exerted positive inhibitory activity, with IC_50_ values of 3.9 μM, 4.3 μM, and 2.7 μM, respectively.

Besides the abovementioned PTP1B inhibitory effects, bromophenols have also been reported to have strong α-glucosidase enzyme inhibitory effects. In fact, α-glucosidase enzyme plays a crucial role in carbohydrates digestion and is a favorite target for anti-diabetic drugs, especially in the case of postprandial hyperglycemia. It has been suggested that Bis-(2,3-dibromo-4,5-di hydroxybenzyl) ether (compound **3**) together with bis(2,3,6-tribromo-4,5-dihydroxybenzyl) ether represents a future kind of α-glucosidase inhibitors [47,48,56,57]. Bis-(2,3,6-tribromo-4,5-dihydroxybenzyl) ether exhibited the most potent α-glucosidase inhibition activity compared to the present series of bromophenols, having an IC_50_ value as low as 0.03 μM [47]. On the other hand, the lowest activity was exhibited by 2,4-dibromophenol (IC_50_ = 110.4 μM). This may bind α-glucosidase inhibition activity to the bromination degree of these metabolites based on the inversely proportional relationship between their IC_50_ value and the number of bromines in metabolites. Likewise, its enzymatic inhibition activity increases proportionately with the increase in the number of phenyl units. Thus, Bis-(2,3-dibromo-4,5-dihydroxybenzyl) ether and Bis-(2,3,6-tribromo-4,5-dihydroxybenzyl) ether, with a diphenyl unit, are found to exhibit much more activity than metabolites with a single phenyl unit, such as 3-bromo-4,5-dihydroxybenzyl alcohol. Nevertheless, the underlying reason behind these changes in biological activity needs further clarification.

Interestingly, along with PTP1B and α-glucosidase inhibition activity, some bromophenols have also shown aldose reductase inhibitory effects. Indeed, aldose reductase is considered the basic enzyme of the polyol pathway, which controls sorbitol formation from glucose and plays a significant role in degenerative complications resulting from diabetes development [58]. 2,2′,3,6,6′-pentabromo-3′,4,4′,5-tetrahydroxydibenzyl ether, Bis-(2,3,6-tribromo- 4,5-dihydroxyphenyl) methane, 2′,3,5′,6-pentabromo-3′,4,4′,5-tetrahydroxydiphenyl methane, 2,3,6-tribromo-4,5-dihydro xymethyl benzene, and 2,3,6-tribromo-4,5-dihydroxybenzaldehyde isolated from the red algae *S. latiuscula* have been reported as aldose reductase inhibitors**,** and thus could be beneficial in T2DM complications (such as eye and nerve damage management) [59].

#### 3.1.2. Brominated Metabolites

In 2010, Qin et al. [60] assessed the in vitro PTP1B inhibitory action of two brominated metabolites (compounds **17** and **18**) isolated from the red algae *Laurencia similis.* Both compounds showed PTP1B inhibition with IC_50_ values of 3.0 μM and 2.7 μM, respectively. The authors also studied the effect of highly brominated metabolites (compounds **19**–**23**), but their corresponding IC_50_ values were much higher than those of the brominated metabolites, ranging from 65.3 µg/mL to 102 µg/mL. This could question the hypothesis that the bromination degree affects the PTP1B inhibitory effect in a directly proportional manner (Table 2).

#### 3.1.3. Polybromodiphenyl Ether Derivatives

Besides the two abovementioned bromophenols (compounds **12** and **13**), the Indonesian marine sponge *Lamellodysidea herbacea* contains six polybromodiphenyl ether derivatives (compounds **24**–**29**). As stated by Yamazaki et al. [45], all of these compounds exhibited in vitro PTP1B inhibitory action with IC_50_ values ranging from 0.6 μM to 1.7 μM. The authors also determined the activity of compounds **24** and **25** in Huh-7, a well-differentiated hepatocyte-derived cellular carcinoma cell line that has been increasingly investigated, given its ability to secrete mitogen hepatoma-derived growth factor, responsible for promoting cell growth without depending on other growth factors present in serum. The IC_50_ values obtained for these compounds were, respectively, 32 µM and 48 µM (Table 3).

#### 3.1.4. Phlorotannins

Back in 1977, Glombitza was the first to introduce the term phlorotannins [61]. Briefly, they are a characteristic type of integral tannins found in brown algae, *Alariaceae*, and are basically classified into six main subclasses (Table 4): eckols, fucols, phlorethols, fucophloretols, fuhalols, and isofuhalos [62,63]. Phlorotannins are thought to modulate cellular signaling, leading to the regulation of different body conditions [49]. Eckol and its derivatives (compounds **30**–**35**) isolated from the edible brown algae *Ecklonia stolonifera* and *Eisenia bicyclis* were studied by several authors [64,65,66], who found variable anti-diabetic effects. The in vitro PTP1B inhibitory activity was variable, with IC_50_ values ranging from 0.6 μM to 55.5 μM, whereas the in vitro α-glucosidase inhibitory action exhibited more potent IC_50_ values ranging from 1.4 μM to 141.2 μM.

It is worth mentioning that phlorofurofucoeckol-A (compound **31**) showed the lowest IC_50_ values for both enzymes. In contrast, phloroglucinol (compound **34**), which is actually the building unit of other polymer phlorotannins, had the highest IC_50_ values for both enzymes. From this, it is possible to suppose that the activity is due to the product of polymerization, not to the basic monomeric structure.

#### 3.1.5. Sterols

This class of secondary metabolites deserves more attention (Table 5). The few studies performed in this field [67,68] focused on hydroperoxyl sterols, epidioxy sterols, and fucosterol contents in different marine invertebrates, including sea sponge *Xestospongia testudinaria* Lamarck (Petrosiidae) from South China, sea urchin *Glyptocidaris crenularis* A. Agassiz (Glyptocidaridae), sea sponge *Mycale* species (Mycalidae), Gorgonian *Dichotella gemmacea* Milne Edwards and Haime (Ellisellidae), algae, seaweed, and diatoms. Several sterols (compounds **36**–**42**) were identified in these different marine species, but their PTP1B inhibitory activity is still undetermined, with the exception of compound **37** (29-hydroperoxystigmasta-5,24(28)-dien-3-ol), in which a positive effect was stated (IC_50_ = 5.8 µg/mL), and compound **42**, which showed moderate PTP1B inhibitory activity [69]. These findings encourage further studies on similar compounds.

#### 3.1.6. Terpenes

##### Sesquiterpenes

Sesquiterpene quinones (compounds **43** and **44**) isolated from the sea sponge *Dysidea* species, available in South China, were evaluated by several research groups [46,70,71] with regards to its in vitro PTP1B inhibitory activity. These sesquiterpenes exhibit positive PTP1B inhibitory activity, with IC_50_ values of 6.7 μM and 9.98 μM, respectively (Table 6). Dehydroeuryspongin A (compound **45**), another sesquiterpene isolated from the marine sponge *Euryspongia* species, also displayed positive activity with an IC_50_ value of 3.6 μM [72].

##### Diterpenes

Diterpenes isolated from *Sarcophyton trocheliophorum* Marenzeller, a Hainan soft coral, received pivotal attention from Liang et al. over two consecutive years [73,74]. The authors isolated three compounds (compounds **46**–**48**), and assessed their in vitro PTP1B inhibitory potential. These diterpenes showed variable effects, with IC_50_ values ranging from 6.8 μM to 27.1 μM.

##### Sesterterpenes

Piao et al. [75] evaluated the activity of two sesterterpenoids (compounds **49** and **50**) from the sponge *Hippospongia lachne* found on Yongxing Island against PTP1B through an in vitro study, and found that they exhibited IC_50_ values of 5.2 μM and 8.7 μM, respectively.

##### Triterpenes

Compound **51**, a triterpene isolated from the Antarctic lichen *Lecidella carpathica*, exhibited prominent in vitro anti-diabetic activity through PTP family enzymes inhibition [76]. Specifically, it inhibited PTP1B (IC_50_ = 3.7 μM) and T-cell protein tyrosine phosphatase (TCPTP) (IC_50_ = 8.4 μM) enzymes. On the other hand, it showed higher IC_50_ values (exceeding 68 μM) against the studied phosphatase enzymes, Src homology phosphatase-2 (SHP-2), leukocyte antigen-related phosphatase (LAR), and protein tyrosine phosphatase receptor type C (PTPRC), also known as CD45 antigen tyrosine phosphatase (Table 6).

Stellettin G (compound **52**), is an isomalabaricane triterpene that was isolated from the Hainan sponge *Stelletta* species by Xue et al. [77]. This compound also displayed prominent in vitro PTP1B inhibitory action (IC_50_ = 4.1 μM). Indeed, isomalabaricanes are currently gaining significant interest as many of them have shown promissory in vitro cytotoxic effects [78].

#### 3.1.7. Fungal Metabolites

In 2013, Sohn et al. and Lee et al. [79,80] investigated the fungal strains derived from marine resources (particularly from *Penicillium* and *Eurotium* species) and isolated seven compounds (compounds **53**–**59**). Compounds **53**–**57** (Table 7) displayed moderate anti-diabetic effects as PTP1B inhibitors, with IC_50_ values varying from 10.7 µM to 64.0 µM, while compounds **58** and **59** showed much lower IC_50_ values (5.3 μM and 1.9 μM).

Aquastatin A (compound 60) is also a fungal metabolite isolated from *Cosmospora* species (Table 7) that has received particular attention from several researchers [81,82]. This attention is attributed not only to its low IC_50_ value against PTP1B enzyme (0.2 μM), but also due to its selective inhibitory activity against others PTPs, including TCPTP, SHP-2, LAR, and CD45.

#### 3.1.8. Miscellaneous Compounds

Fucoxanthin, compound **61**, is a carotenoid that was isolated from *Phaeodactylum tricornutum* and edible brown seaweeds, such as *Eisenia bicyclis* (Arame), *Undaria pinnatifida* (Wakame), and *Hi-jikia fusiformis* (Hijiki) (Table 8). Fucoxanthin was found to decrease blood glucose and insulin levels as well as water intake in a diabetic/obese KKAy mice model. A significant reduction in mRNA expression levels of monocyte chemoattractant protein-1 and tumor necrosis factor-α (TNF-α) was observed, which is believed to be involved in insulin resistance induction. A prominent PTP1B inhibitory action was also stated, with an IC_50_ value of 4.8 μM [83,84,85].

Seo et al. [76] isolated brialmontin 1 and atraric acid (compounds **62** and **63**) from *Lecidella carpathica*, an Antarctic lichen. Both compounds were found to have PTP1B inhibitory action with IC_50_ values of 14 μM and 51.5 μM, respectively (Table 8).

In addition, the crude extracts from other seaweeds, including *Derbesia marina*, *Symphyocladia latiuscula*, *Codium adhaerens*, *Attheya longicornis*, *Chaetoceros socialis*, *Chaetoceros furcellatus*, *Skeletonema marinoi*, and *Porosira glacialis* along with the brown algae *Hisikia fuziformis* have been also evaluated for their in vitro PTP1B inhibitory capacity [86,87,88].

Furthermore, the ethanolic extract of the brown algae *S. serratifolium* C. Agardh exhibited broad PTP1B and α-glucosidase inhibitory activities (IC_50_ 7.04 and 24.16 µg/mL, respectively) [89]. Among the four subfractions of the ethanol extract, n-hexane exhibited the highest activities (IC50 1.88 and 3.16 µg/mL, respectively), so its major three plastoquinones—sargahydroquinoic acid, sargachromenol, and sargaquinoic acid (compounds **64**–**66**)—were isolated and the three compounds showed potent PTP1B inhibitory activity (Table 8). Sargachromenol also showed the most promising α-glucosidase inhibitory activity with an IC_50_ value of 42.41 μM, followed by sargaquinoic acid with an IC_50_ value of 96.17 ± 3.48 μM, while sargahydroquinoic acid was inactive.

### 3.2. PTP1B Inhibitory Activity: In Vivo Findings

With regards to the in vivo anti-diabetic activity of marine plants, Shi et al. [53] assessed the in vivo PTP1B inhibitory activity of highly brominated derivatives isolated from the red algae *R. confervoides*, which contains one or two 2,3-dibromo-4,5-dihydroxybenzyl units, in diabetic rats. They found that *R. confervoides* extracts led to a significant reduction in serum glucose levels. These in vivo results may confirm that the anti-hyperglycemic activity of *R. confervoides* is partially attributable to the PTP1B inhibition activity of its constituents [53]. Similarly, Nuño et al. [90] investigated the anti-diabetic activities of the microalgae haptophyte *Isochrysis galbana* and the ochrophyte *Nannochloropsis oculata* in a diabetic rat model. Different biochemical parameters were investigated, such as glucose level, body weight, lipoproteins, and nitrogenous compounds. In addition, gastrointestinal (GI) histopathology was studied. Both microalgae studied led to an increase in low-density lipoprotein (LDL) and a decrease in high-density lipoprotein (HDL) levels in both control and diabetic rats. More specifically, *I. galbana* decreased body weight, glucose, triacylglycerides, and cholesterol levels and exhibited just slight signs of inflammation in the gut. The observed activity could be attributed to their high content in docosahexaenoic (DHA) and eicosapentaenoic (EPA) fatty acids. The *N. oculata*-treated diabetic group did not show any changes in clinical values and had negative effects within the GI tract. Further studies are needed to confirm the effective employment of *I. galbana* as an anti-diabetic functional food.

## 4. Current Trends and Upcoming Challenges

### 4.1. In Vitro and In Vivo Concerns

In overall, T2DM is a metabolic disease characterized by hyperglycemia and hyperinsulinemia, in which the most common risk factor is overweight or obesity [91]. The development of diabetes may occur as a result of insulin secretion and/or signaling deregulation by insulin receptors (IR) [92]. The action of PTPs on IR themselves or their substrates is an important mechanism in insulin signaling regulation [92]. The role of PTPs in insulin signaling pathways and diabetes has previously been studied using vanadium compounds, which are able to reduce serum glucose levels in both type 1 and type 2 diabetic animal models [93,94]. Vanadium compounds show fundamental in vitro and in vivo insulinomimetic effects. Thus, the oral administration of such compounds promotes the normalization of serum glucose levels in T2DM rats, increasing glucose uptake [95]. These rats exhibited increased levels of hepatic cytosolic PTP activity, which decreased following insulin and vanadate treatment, leading to serum glucose levels normalization. These findings can be explained through PTPs inhibition with consequent improvement of cellular tyrosine phosphorylation [96].

Besides IR recognition, structural studies on PTP1B enzymes led to the identification of JAK2 and tyrosine kinase 2 (TYK2) as potential PTP1B substrates. Following interferon stimulation, both kinases were found to be hyperphosphorylated in PTP1B null fibroblasts [97]. This finding was further confirmed by the negative regulation of leptin-stimulated JAK2 phosphorylation produced by PTP1B, which reduced leptin signaling in in vivo models. Thus, null PTP1B mutation was introduced into leptin-deficient obese *ob/ob* mice, and a significant decrease in weight gain with an increase in resting metabolic rates was found in PTP1B-deficient *ob/ob* mice. Moreover, fat pads analysis proposed that the weight variations could be attributed to a decrease in adipose tissue. So, in the absence of leptin, PTP1B loss can reduce weight gain without modifying food intake [98,99]. Moreover, PTP1B-deficient mice had an increased response to weight loss caused by leptin and feeding suppression. The hypothalami of these mice evidenced a marked improvement in leptin-induced Transcription Factor STAT3 phosphorylation, indicating that the introduction of exogenous leptin in PTP1B deficiency will also led to leptin sensitivity enhancement [98,99]. Actually, substrate trapping trials using catalytically inactive PTP1B D181A confirmed that leptin-activated JAK2 is considered a PTP1B substrate, and that leptin signaling reduction is an obesity resistance mechanism in PTP1B null mice.

### 4.2. Human Concerns

In humans, weight loss and improved insulin sensitivity are closely related to decreased PTP activity, along with LAR and PTP1B expression in adipose tissue [100]. It is noticeable that PTP1B activity is not always related to its level of expression.

In obese and diabetic subjects, PTP1B protein levels show a 3- to 5-fold increase in abdominal adipose tissue, while a notable decrease in PTP1B activity was observed per unit of PTP1B protein [101]. It was also observed that total cellular PTP, not increased PTP1B activity, led to a marked raise in adipose tissue in obese individuals. In addition, it has been reported that an increase in PTP activity, but not in PTP1B activity, is accompanied by reduced insulin-stimulated glucose transport, proposing a tissue-specific role in glucose homeostasis for PTP1B [102].

On the other hand, genetic evidence also links PTP1B to diabetes and obesity in humans. Indeed, PTP1B locus maps to chromosome 20 in the region q13.1–q13.2 [103], which is a region recognized as a quantitative trait locus linked to insulin and obesity. A correlation between the role of PTP1B in insulin resistance and various polymorphisms has also been reported. That is, there is a continuing need to identify new PTP inhibitors for diabetes and obesity control.

### 4.3. Concerns in Culture Conditions

The anti-diabetic activity of several microalgae cultured under stressful conditions was also assessed using PTP1B assay [86,87]. Ingebrigtsen et al. [86] evaluated the non-polar fraction of five diatoms isolated from the North Atlantic (i.e., *Attheya longicornis, Chaetoceros socialis, Chaetoceros furcellatus, Skeletonema marinoi*, and *Porosira glacialis*), grown under variable light/temperature conditions. *A. longicornis* and *C. furcellatus* extracts displayed anti-PTP1B activity. On the other hand, *C. socialis* showed activity only when grown under high temperature—low light conditions, whereas *P. glacialis* showed activity when cultivated only under high temperature—high light conditions. Still, *S. marinoi* showed no activity in any of the studied conditions. Thus, these findings highlight the importance of culture conditions in activating bioactive metabolites production.

Meanwhile, Lauritano et al. [87] screened a total of 32 crude extracts from microalgal species (four flagellates, seven dinoflagellates, and 21 diatoms) grown under variable culturing conditions. It was found that temperature/light stresses are more important than nutritional stress in microalgal species containing bioactive metabolites that have PTP1B enzyme inhibition.

## 5. Conclusions

In general, the data collected emphasize the importance of chemical constituents from different marine species, given their PTP1B inhibition activity, as key targets in T2DM and obesity management.

Although distinct anti-diabetic therapeutic strategies are currently available, there is still an urgent need to find more effective and less toxic pharmacological agents. PTP1B has been established as a promising molecular target for the treatment of both T2DM and its major risk factor, obesity. However, PTP1B inhibitors still present significant problems over the closely related enzymes belonging to the PTP family, due to their poor selectivity, which is the main point to overcome their adverse effects.

As summarized herein, marine metabolites have recently gained significant attention from the scientific community, as they are considered to represent a repository of diverse unexploited bioactivities and structural features that could broaden the chemical library and may provide potential targets for the discovery of novel PTP1B inhibitory agents. However, another challenge must be kept in mind, arising from the limited yields of marine metabolites, which hinder their assessment through in vivo studies. In addition, it should be noted that current studies mainly focus on marine metabolites isolation and characterization as PTP1B inhibitors. However, the underlying mechanisms of action and structure–activity relationships need more attention. Thus, intensive efforts should be made through high-throughput screening of marine metabolites, along with structural optimization and synthesis of new PTP1B inhibitors, in order to identify selective, safer, and more effective PTP1B inhibitory agents in the near future.

## Figures and Tables

**Figure 1 molecules-23-03334-f001:**
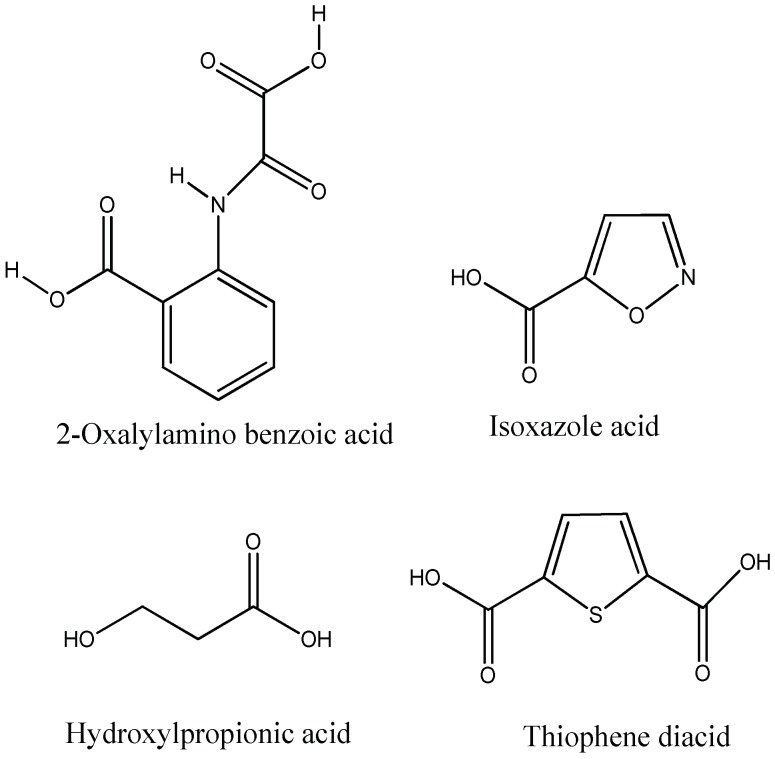
Structures of phosphotyrosine (pTyr) surrogate acids.

**Table 1 molecules-23-03334-t001:** Marine plant-isolated bromophenols with in vitro PTP1B inhibitory effects.

No.	Compound/Structure	Marine Species	Outcomes/Enzymes	Reference
1	2,2′,3,3′-Tetrabromo-4,4′,5,5′-tetra-hydroxydiphenyl methane 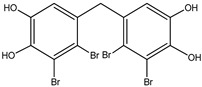	Red algae *Rhodomela confervoides*	PTP1B inhibition (IC_50_ = 2.4 μM)	[53]
2	3-Bromo-4,5-Bis-(2,3-dibromo-4,5-dihydroxybenzyl)pyrocatechol 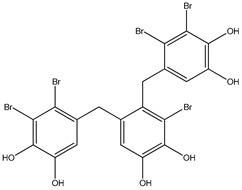	Red algae *Rhodomela confervoides*	PTP1B inhibition (IC_50_ = 1.7 μM)	[53]
3	Bis-(2,3-dibromo-4,5-dihydroxybenzyl) ether 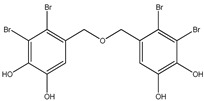	Red algae *Rhodomela confervoides*	PTP1B inhibition (IC_50_ = 1.5 μM) α-glucosidase inhibition (IC_50_ = 0.098 μM)	[53]
4	2,2′,3,3′-Tetrabromo-3′,4,4′,5-tetrahydroxy-6′-ethyloxymethyldiphenylmethane 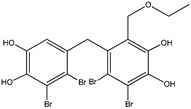	Red algae *Rhodomela confervoides*	PTP1B inhibition (IC_50_ = 0.8 μM)	[53]
5	3,4-Dibromo-5-(2-bromo-3,4-dihydroxy-6-(ethoxymethyl)benzyl)benzene-1,2-diol 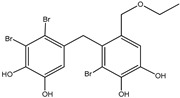	Red algae *Rhodomela confervoides*	PTP1B inhibition (IC_50_ = 0.8 μM)	[49,50,51,52]
6	3,4-Dibromo-5-(methoxymethyl)benzene-1,2-diol 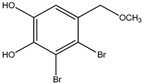	Red algae *Rhodomela confervoides*	PTP1B inhibition (IC_50_ = 3.4 μM)	[51,53]
7	3-(2,3-Dibromo-4,5-dihydroxyphenyl)-2-methylpropanal 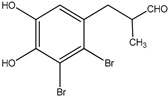	Red algae *Rhodomela confervoides*	PTP1B inhibition (IC_50_ = 4.5 μM)	[51,53]
8	3,4-Dibromo-5-(2-bromo-3,4-dihydroxy-6-(isobutoxymethyl)benzyl)benzene-1,2-diol 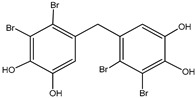	Red algae *Rhodomela confervoides*	PTP1B inhibition (IC_50_ = 2.4 μM)	[51,53]
9	7-Bromo-1-(2,3-dibromo-4,5-dihydroxy phenyl)-2,3-dihydro-1H-indene-5,6-diol 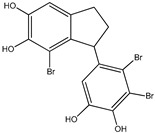	Red algae *Rhodomela confervoides*	PTP1B inhibition (IC_50_ = 2.8 μM)	[51,53]
10	5,5’-(3-Bromo-4,5-dihydroxy-1,2-phenylene)-Bis-(methylene))Bis-(3,4-dibromobenzene-1,2-diol) 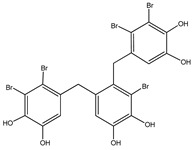	Red algae *Rhodomela confervoides*	PTP1B inhibition (IC_50_ = 1.7 μM)	[51,53]
11	3,4-Dibromo-5-(2-bromo-3,4-dihydroxy-6-(ethoxymethyl)benzyl)benzene-1,2-diol 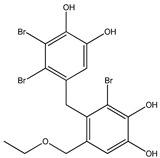	Red algae *Rhodomela confervoides*	PTP1B inhibition (IC_50_ = 0.84 μM)	[50,51]
12	2-(3′,5′-Dibromo-2′-methoxyphenoxy)-3,5-dibromophenol 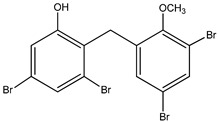	Indonesian marine sponge *Lamellodysidea herbacea*	PTP1B inhibition (IC_50_ = 0.9 μM)	[45]
13	2-(3′,5′-Dibromo-2′-methoxyphenoxy)-3,5-dibromophenol-methyl ether 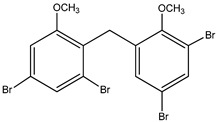	Indonesian marine sponge *Lamellodysidea herbacea*	PTP1B inhibition (IC_50_ = 1.7 μM)	[45]
14	2,3,6-Tribromo-4,5-dihydroxybenzyl methyl ether 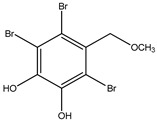	Marine red algae *Symphyocladia latiuscula*	PTP1B inhibition (IC_50_ = 3.9 µM)	[55]
15	Bis-(2,3,6-tribromo-4,5-dihydroxyphenyl) methane 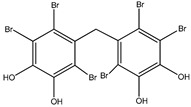	Marine red algae *Symphyocladia latiuscula*	PTP1B inhibition (IC_50_ = 4.3 µM)	[55]
16	1,2-Bis-(2,3,6-tribromo-4,5-dihydroxyphenyl)-ethane 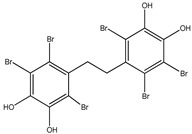	Marine red algae *Symphyocladia latiuscula*	PTP1B inhibition (IC_50_ = 2.7 µM)	[55]

**Table 2 molecules-23-03334-t002:** Marine plant-isolated brominated metabolites with in vitro PTP1B inhibitory effects.

No.	Compound/Structure	Marine Species	Outcomes/Enzymes	Reference
17	3’,5’,6’,6-Tetrabromo-2,4-dimethyldiphenyl ether 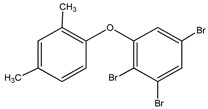	Red algae *Laurencia similis*	PTP1B inhibition (IC_50_ = 3.0 μM)	[60]
18	2’,5’,6’,5,6-Pentabromo-3’,4’,3,4-tetramethoxybenzo-phenone 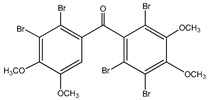	Red algae *Laurencia similis*	PTP1B inhibition (IC_50_ = 2.7 μM)	[60]
19	3’,5’,6’6-Tetrabromo-2,4-dimethyldiphenyl ether 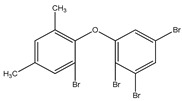	Red algae *Laurencia similis*	PTP1B inhibition (IC_50_ = 3.0 µg/mL)	[60]
20	1,2,5-Tribromo-3-bromoamino-7-bromomethylnaphthalene 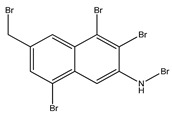	Red algae *Laurencia similis*	PTP1B inhibition (IC_50_ = 102 µg/mL)	[60]
21	2,5,8-Tribromo-3-bromoamino-7-bromomethylnaphthalene 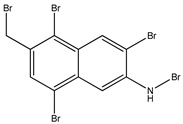	Red algae *Laurencia similis*	PTP1B inhibition (IC_50_ = 65.3 µg/mL)	[60]
22	2,5,6-Tribromo-3-bromoamino-7-bromomethylnaphthalene 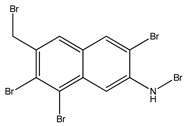	Red algae *Laurencia similis*	PTP1B inhibition (IC_50_ = 69.8 µg/mL)	[60]
23	2’,5’,6’,5,6-Pentabromo-3’,4’,3,4-tetramethoxybenzo-phenone 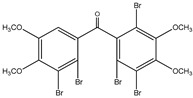	Red algae *Laurencia similis*	PTP1B inhibition (IC_50_ = 2.7 µg/mL)	[60]

**Table 3 molecules-23-03334-t003:** Marine plant-isolated polybromodiphenyl ether derivatives with in vitro PTP1B inhibitory effects.

No.	Compound/Structure	Marine Species	Outcomes/Enzymes	Reference
24	2-(3’,5’-Dibromo-2’-methoxyphenoxy)-3,5-dibromophenol 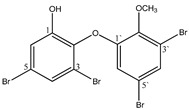	Indonesian marine sponge *Lamellodysidea herbacea*	PTP1B inhibition (IC_50_ = 0.9 µM) Huh-7 inhibition (IC_50_ = 32 µM)	[45]
25	3,5-Dibromo-2-(3’,5’-dibromo-2’-methoxyphenoxy)-1-methoxybenzene 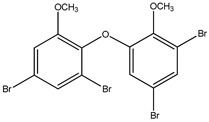	Indonesian marine sponge *Lamellodysidea herbacea*	PTP1B inhibition (IC_50_ = 1.7 µM) Huh-7 inhibition (IC_50_ = 48 µM)	[45]
26	3,5-Dibromo-2-(3’,5’-dibromo-2’ -methoxyphenoxy)phenylethanoate 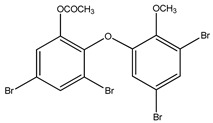	Indonesian marine sponge *Lamellodysidea herbacea*	PTP1B inhibition (IC_50_ = 0.6 µM)	[45]
27	3,5-Dibromo-2-(3’,5’-dibromo-2’ -methoxyphenoxy)phenylbutanoate 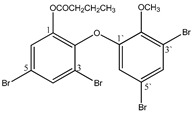	Indonesian marine sponge *Lamellodysidea herbacea*	PTP1B inhibition (IC_50_ = 0.7 µM)	[45]
28	3,5-Dibromo-2-(3’,5’-dibromo-2’ -methoxyphenoxy)phenylhexanoate 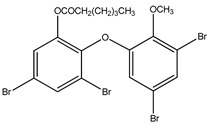	Indonesian marine sponge *Lamellodysidea herbacea*	PTP1B inhibition (IC_50_ = 0.7 µM)	[45]
29	3,5-Dibromo-2-(3’,5’-dibromo-2’ -methoxyphenoxy)phenyl benzoate 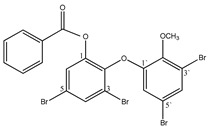	Indonesian marine sponge *Lamellodysidea herbacea*	PTP1B inhibition (IC_50_ = 1.0 µM)	[45]

**Table 4 molecules-23-03334-t004:** Marine plant-isolated phlorotannins with in vitro PTP1B inhibitory effects.

No.	Compound/Structure	Marine Species	Outcomes/Enzymes	Reference
30	Eckol 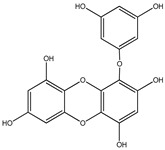	Edible brown algae *Ecklonia stolonifera* and *Eisenia bicyclis*	PTP1B inhibition (IC_50_ = 2.6 µM) α-glucosidase inhibition (IC_50_ = 22.8 µM)	[64,65]
31	Phlorofurofucoeckol-A 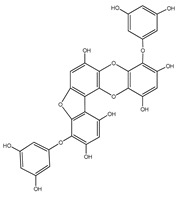	Edible brown algae *Ecklonia stolonifera* and *Eisenia bicyclis*	PTP1B inhibition (IC_50_ = 0.6 µM) α-glucosidase inhibition (IC_50_ = 1.4 µM)	[64,65]
32	Dieckol 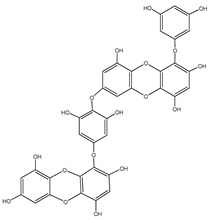	Edible brown algae *Ecklonia stolonifera* and *Eisenia bicyclis*	PTP1B inhibition (IC_50_ = 1.2 µM) α-glucosidase inhibition (IC_50_ = 1.6 µM)	[64,65]
33	7-Phloroeckol 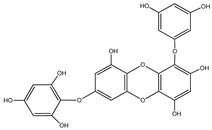	Edible brown algae *Ecklonia stolonifera* and *Eisenia bicyclis*	PTP1B inhibition (IC_50_ = 2.1 µM) α-glucosidase inhibition (IC_50_ = 6.1 µM)	[65]
34	Phloroglucinol 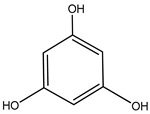	Edible brown algae *Ecklonia stolonifera* and *Eisenia bicyclis*	PTP1B inhibition (IC_50_ = 55.5 µM) α-glucosidase inhibition (IC_50_ = 141.2 µM)	[65]
35	Dioxinodehydroeckol 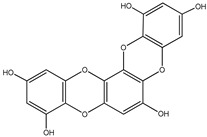	Edible brown algae *Ecklonia stolonifera* and *Eisenia bicyclis*	PTP1B inhibition (IC_50_ = 30.0 µM) α-glucosidase inhibition (IC_50_ = 34.6 µM)	[65]

**Table 5 molecules-23-03334-t005:** Marine plant-isolated sterols with in vitro PTP1B inhibitory effects.

No.	Compound/Structure	Marine Species	Outcomes/Enzymes	Reference
36	24-Hydroperoxy-24-Vinylcholesterol 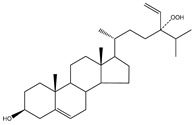	Marine invertebrates South China Sea sponge *Xestospongia testudinaria* Lamarck (Petrosiidae)	-	[68]
37	29-Hydroperoxystigmasta-5,24(28)-dien-3-ol 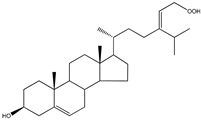	Marine invertebrates South China Sea sponge *Xestospongia testudinaria* Lamarck (Petrosiidae)	PTP1B inhibition (IC_50_ = 5.8 µg/mL)	[68]
38	5α,8α-Epidioxycholest-6-en-3β-ol 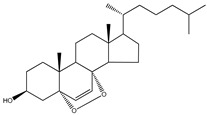	Sea urchin *Glyptocidaris crenularis* A. Agassiz (Glyptocidaridae)	-	[68]
39	5α,8α-Epidioxycholest-6,22-dien-3β-ol 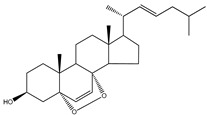	Sponge *Mycale* spp. (Mycalidae)	-	[68]
40	5α,8α-Epidioxy-ergosta-6,22-dien-3β-ol 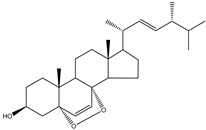	Gorgonian *Dichotella gemmacea* Milne Edwards and Haime (Ellisellidae)	-	[68]
41	3β-Hydroxycholest-5-en-25-acetoxy-19-oate 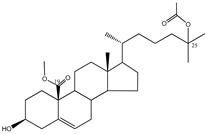	Gorgonian *Dichotella gemmacea* Milne Edwards and Haime (Ellisellidae)	-	[68]
42	Fucosterol (24-ethylidene cholesterol) 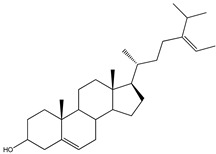	Brown algae *Eisenia bicyclis* and *Ecklonia* *stolonifera*	Non-competitive type inhibitor against PTP1B	[67,68,69]

**Table 6 molecules-23-03334-t006:** Marine plant-isolated terpenes with in vitro PTP1B inhibitory effects.

No.	Compound/Structure	Marine Species	Outcomes/Enzymes	Reference
43	Dysidine 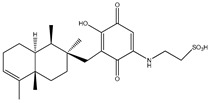	Sponge *Dysidea villosa*	PTP1B inhibition (IC_50_ = 6.7 μM)	[46,71]
44	Dysidavarone A 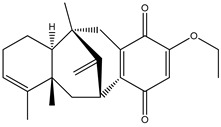	South China Sea sponge *Dysidea avara*	PTP1B inhibition (IC_50_ = 10.0 μM)	[70]
45	Dehydroeuryspongin A 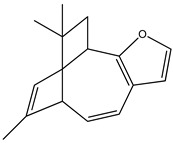	Marine Sponge *Euryspongia* spp.	PTP1B inhibition (IC_50_ = 3.6 μM)	[72]
46	Sarsolilide A 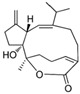	Hainan soft coral *Sarcophyton trocheliophorum* Marenzeller	PTP1B inhibition (IC_50_ = 6.8 μM)	[73]
47	Sarsolilide B 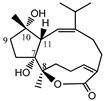	Hainan soft coral *Sarcophyton trocheliophorum* Marenzeller	PTP1B inhibition (IC_50_ = 27.1 μM)	[73]
48	Methyl sarcotroates A and B 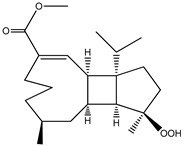	Hainan soft coral *Sarcophyton trocheliophorum*	PTP1B inhibition (IC_50_ = 7.0 μM)	[73]
49	9-Oxa-2-azabicyclo-[3,3,1]-nona-3,7-diene derivative 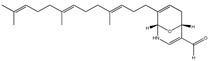	Sponge *Hippospongia lachne* of Yongxing Island	PTP1B inhibition (IC_50_ = 5.2 μM)	[75]
50	2-(Aminomethylene) hepta-3,5-dienedial moiety connected with farnesyl group at C-7 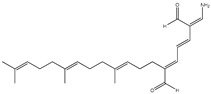	Sponge *Hippospongia lachne* of Yongxing Island	PTP1B inhibition (IC_50_ = 8.7 μM)	[75]
51	Hopane-6*α*,22-diol 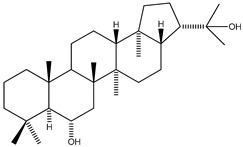	Antarctic lichen *Lecidella carpathica*	PTP1B inhibition (IC_50_ =3.7 μM)TCPTP inhibition (IC_50_ = 8.4 μM)SHP-2 inhibition (IC_50_ *>* 68 μM)LAR inhibition(IC_50_ *>* 68 μM)CD45 inhibition (IC_50_ *>* 68 μM)	[76]
52	Stellettin G 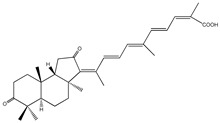	Hainan sponge *Stelletta* spp.	PTP1B inhibition (IC_50_ = 4.1 μM)	[77]

TCPTP, T-cell protein tyrosine phosphatase; SHP-2, src homology phosphatase-2; LAR, leukocyte antigen-related phosphatase; CD45, CD45 tyrosine phosphatase.

**Table 7 molecules-23-03334-t007:** Marine plant-isolated fungal metabolites with in vitro PTP1B inhibitory effects.

No.	Compound/Structure	Marine Species	Outcomes/Enzymes	Reference
53	Fructigenine A 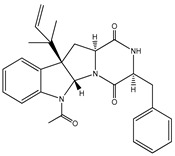	Marine-derived fungal strains *Penicillium* and *Eurotium* species	PTP1B inhibition (IC_50_ = 10.7 µM)	[80]
54	Cyclopenol 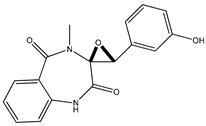	Marine-derived fungal strains *Penicillium* and *Eurotium* species	PTP1B inhibition (IC_50_ = 30.0 µM)	[80]
55	Echinulin 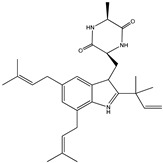	Marine-derived fungal strains *Penicillium* and *Eurotium* species	PTP1B inhibition (IC_50_ = 29.4 µM)	[80]
56	Flavoglaucin 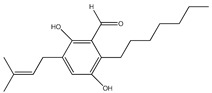	Marine-derived fungal strains *Penicillium* and *Eurotium* species	PTP1B inhibition (IC_50_ = 13.4 µM)	[80]
57	Viridicatol 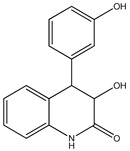	Marine-derived fungal strains *Penicillium* and *Eurotium* species	PTP1B inhibition (IC_50_ = 64.0 µM)	[80]
58	Penstyrylpyrone 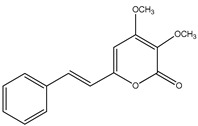	Marine-derived fungi *Penicillium* JF-55 cultures	PTP1B inhibition (IC_50_ = 5.3 μM)	[82]
59	Anhydrofulvic acid 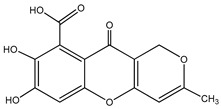	Marine-derived fungi *Penicillium* JF-55 cultures	PTP1B inhibition (IC_50_ = 1.9 μM)	[82]
60	Aquastatin A 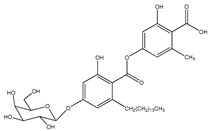	Fungus *Cosmospora* species	PTP1B inhibition (IC_50_ = 0.2 μM), as well as inhibition of TCPTP, SHP-2, LAR, and CD45 activity	[81,82]

**Table 8 molecules-23-03334-t008:** Marine plant-isolated miscellaneous compounds with in vitro PTP1B inhibitory effects.

No.	Compound/Structure	Marine Species	Outcomes/Enzymes	Reference
61	Fucoxanthin 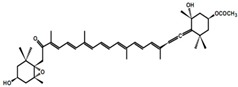	*Phaeodactylum tricornutum, Eisenia bicyclis* (Arame), *Undaria pinnatifida* (Wakame), and *Hi-jikia fusiformis* (Hijiki)	PTP1B inhibition (IC_50_ = 4.8 μM)	[83,84,85]
62	Brialmontin 1 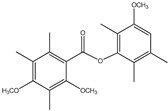	Antarctic lichen *Lecidella carpathica*	PTP1B inhibition (IC_50_ =14 μM)	[76]
63	Atraric acid 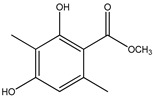	Antarctic lichen *Lecidella carpathica*	PTP1B inhibition (IC_50_ = 51.5 μM)	[76]
64	Saragahydroquinoic acid 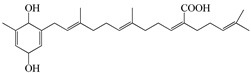	Brown algae *Sargassum serratifolium* C. Agardh	PTP1B inhibition (IC_50_ = 5.14 μM)	[89]
65	Saragaquinoic acid 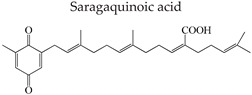	Brown algae *Sargassum serratifolium* C. Agardh	PTP1B inhibition (IC_50_ = 14.15 μM)	[89]
66	Sargachromenol 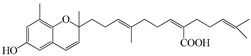	Brown algae *Sargassum serratifolium* C. Agardh	PTP1B inhibition (IC_50_ = 11.80 μM)	[89]

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
