# Peer review of "Looking at Marine-Derived Bioactive Molecules as Upcoming Anti-Diabetic Agents: A Special Emphasis on PTP1B Inhibitors"

_molecules, 2018, doi:10.3390/molecules23123334_

Reviewer 1 Report

Dear Editor in Chief,

The manuscript entitled Marine-derived bioactive molecules as upcoming anti-diabetic agents: A reviewhas been reviewed. There are ample of typographical errors throughout the manuscript. And also, the reference formatting of manuscript is very poor. Since, these are the basic things that authors should considered first, I strongly recommend to go through the guideline and correct it thoroughly. I have listed major and minor comments below for author’s convenience:

Major comments:

General:

 From the manuscript, it seems that the authors are mainly trying to focused on the PTP1B inhibition. If so, the title could also be changed to the specific one focusing PTP1B inhibition. Otherwise, if they want to review on anti-diabetic agents, the content for the α-glucosidase and aldose reductase should be reviewed and revised more. As here, they only focused on brominated compounds.

And also, authors mentioned about the α-glucosidase inhibition in PTP1B table and later on using paragraph under the topic “Marine-derived molecules with α-glucosidase inhibition” as well. This can arrange under one topic rather than separately or the tables can be modified.

The authors should review all the articles as some of them looks missing:

For example:

Ali, M.Y., Kim, D.H., Seong, S.H., Kim, H.R., Jung, H.A. and Choi, J.S., 2017. α-Glucosidase and protein tyrosine phosphatase 1B inhibitory activity of plastoquinones from marine brown alga Sargassum serratifolium. Marine Drugs, 15(12), p.368.

Since the authors are comparing the metabolites activity, authors can mention a topic as possible compounds for anti-diabetic activity which mainly include marine isolation having potential of exerting PTP1B and or α-glucosidase inhibition.

For example:

Wagle, A., Seong, S.H., Zhao, B.T., Woo, M.H., Jung, H.A. and Choi, J.S., 2018. Comparative study of selective in vitro and in silico BACE1 inhibitory potential of glycyrrhizin together with its metabolites, 18α-and 18β-glycyrrhetinic acid, isolated from Hizikia fusiformis. Archives of Pharmacal Research, 41(4), pp.409-418.

Introduction:

Line 111, Here in the reference, there are other biological activity included. Is there any reason, authors focused only in these mentioned activity?

Table 5, Compound 42, Fucosterol. The reference mentioned for this compound does not include any information regarding it.

Similar happened in the table 6, The reference for the compound Dysidine (45, 65) does not match with content. The reference should be “Li, Y., Zhang, Y., Shen, X. and Guo, Y.W., 2009. A novel sesquiterpene quinone from Hainan sponge Dysidea villosa. Bioorganic & medicinal chemistry letters, 19(2), pp.390-392”.

Likewise, again in Table 6, the reference author cited does not have information for Dysidavarone A.

Above mentioned are only few listed, however, there are plenty of such major mistakes. I would like to advice to go thoroughly through the manuscript and correct the mistakes.

The references mentioned in the Table 6 does not match with the text. Again, similar in case of Table 7 and 8 also.

In figure 2, the name of the structure is incorrect. (2ʹ,3,5ʹ,6-pentabromo-3ʹ,4,4ʹ,5-tetrahydroxydiphenylmethane)

Minor comments:

There are a lot of grammatical errors in the text. Some of them along with the other errors as well have been mentioned in the following text.

Abstract:

Line 39, “Structural-activity relationships (SARs) that could be drawn from the available data”, not “drown”.

Introduction:

Line 46, The reference provided for the statistical data should be latest one. For example; Reference 1- “The burden of mortality attributable to diabetes: Realistic estimates for the year 2000”. Similar condition applies to the reference 3, line 49.

Line 57, “beneficial in improving the quality life of T2DM patients sound better than beneficial in improving the quality of life of T2DM patients.

Line 57-60, The sentence looks a bit unusual when read, especially after mechanism of action, such as insulin release stimulation……….”. Modify the sentence.

Line 63, Rather than citing an article, it’s better to cite a book for reference 11.

Line 95, “and being hydrophobic in nature” instead of “and being of hydrophobic nature”.

Line 114, “Macroalgae species can reach 9.000 species?” There should be comma instead of fullstop.

Line 132-134, This sentence should be cited with suitable reference. (The isolation and identification…………… possessing PTP1B inhibitory activity).

Line 142, “some of these compounds are discussed” instead of “is discussed”.

In Table 1, Number 11 compound, there is no PTP1B inhibition activity mentioned.

In Table 1, Compounds 14-16, The unit mentioned in the table is different with that of text at line 157-159.

Use same number of significant figures after every decimal place.

The symbol prime -“ ʹ ” is denoted differently in many place. Kindly, go through the manuscript and make it uniform throughout.

Table 2, Compound 23, make it uniform throughought the manuscript. IC50 2.66 µg/mL while others have IC50 = ….

Line 174, There is mentioned six polybromodiphenyl ether derivatives, but in table no any six polybrominated compounds seen.

Line 177, Its better to mention the significance of mentioning Huh-7 inhibition for compounds 24 and 25.

Table 4. In this table, the IC50 value is along with the standard error of mean (2.64 ±0.04 µM), however, others table include only IC50 value. Make it uniform throughout the manuscript. And, there should be space between the … “±0.04”.

What is the significance of the reference article 78? What are the points that are cited from this reference?

In reference 83, regarding PTP1B, author mentioned as, “The potential of microalgae to block PTP1B activity was screened, but no apparent bioactivity was observed (data not shown)”. Here the meaning of the sentence is not clear.

Line 274-277, The sentences and reference cited are quite confusing with each other. And the spelling of Symphyocladia latiuscula is incorrect in the manuscript (Line 274). Since, these are again explained in details in “Concerns in culture media”, here it can be modified mentioning only what is necessary over there.

Line 339, “through using vanadium compounds” sounds quite strange. Better to write only “using”.

Line 342, Is it T1DM rats?

Line 374, Chromosome 20q13.1-q13.2 [102]…., it’s better to write chromosome 20 in the region q13.1-q13.2 or similar to that.

References:

The references are not aligned properly.

Use punctuations (comma, period and semicolon).

Check for the title of the study and make it in uniform pattern (either all words capital or small).

The author names are arranged randomly. Somewhere it is abbreviation and somewhere not. And also no use of proper punctuation and space bar. For example;

Reference 58- M., T. N. M. a. A. T ???

Reference 60- Jung A. H., Y. N. Y, Woo M. which actually should be Jung H.A., Yoon N. Y., Woo, M. H and so on.

Reference 63- Abdul QA, C, R., Jung HA, Choi JS. , Heres it should be Choi R. J., and punctuation mark is missing in this reference. Follow same pattern within the manuscript.

Reference 79- the name J.S. is at last after the title of the article.

Reference 86- Kim, K. Y. N., K.A.;…. ????

(These are only some that has been enlisted. There are plenty of such mistakes in almost every references. I strongly recommend to go through the author’s guideline of the manuscript and follow the same pattern in each and every references)

Format for Molecules journal article:

Author 1, A.B.; Author 2, C.D. Title of the article. Abbreviated Journal Name YearVolume, page range. Available online: URL (accessed on Day Month Year).

(Similarly, see the journal website for others formatting).

The issue is mentioned in some places while missing in some. Follow journal guideline and do according. Make it uniform.

The page number range is missing. Check and correct it.

Somewhere there is use of journal abbreviations form while some have full name. Check and correct it.

The scientific name of plants and animals should be in italic. Check thoroughly.

The title is missing in reference 69.

I would suggest you to go along with the journal’s reference guideline. Most of the references are not in journal (Molecules) format. Check and correct.

Remarks: There are plenty of mistakes seen while referencing the manuscript. As a review article, these mistakes can lead to the decline of the true values of review paper. And also, the reference section, plenty of mistakes are seen. I would like to consider the above mentioned comments and make revisions accordingly.

Author Response

Dear Editor in Chief,

The manuscript entitled “Marine-derived bioactive molecules as upcoming anti-diabetic agents: A review” has been reviewed. There are ample of typographical errors throughout the manuscript. And also, the reference formatting of manuscript is very poor. Since, these are the basic things that authors should considered first, I strongly recommend to go through the guideline and correct it thoroughly. I have listed major and minor comments below for author’s convenience:

Answer: Thank you for the reviewer advice. We modified all the required aspects accordingly

Major comments:

General:

From the manuscript, it seems that the authors are mainly trying to focused on the PTP1B inhibition. If so, the title could also be changed to the specific one focusing PTP1B inhibition. Otherwise, if they want to review on anti-diabetic agents, the content for the α-glucosidase and aldose reductase should be reviewed and revised more. As here, they only focused on brominated compounds. And also, authors mentioned about the α-glucosidase inhibition in PTP1B table and later on using paragraph under the topic “Marine-derived molecules with α-glucosidase inhibition” as well. This can arrange under one topic rather than separately or the tables can be modified.

Answer: Title was changed accordingly, and the subtopics Marine-derived molecules with α-glucosidase and aldose reductase inhibition effects were restructured an included in previous section.

The authors should review all the articles as some of them looks missing:

For example: 

Ali, M.Y., Kim, D.H., Seong, S.H., Kim, H.R., Jung, H.A. and Choi, J.S., 2017. α-Glucosidase and protein tyrosine phosphatase 1B inhibitory activity of plastoquinones from marine brown alga Sargassum serratifolium. Marine Drugs, 15(12), p.368.

Answer: This reference and its content was added in subsection: 3.1.8. Miscellaneous compounds

Since the authors are comparing the metabolites activity, authors can mention a topic as possible compounds for anti-diabetic activity which mainly include marine isolation having potential of exerting PTP1B and or α-glucosidase inhibition.

For example: 

Wagle, A., Seong, S.H., Zhao, B.T., Woo, M.H., Jung, H.A. and Choi, J.S., 2018. Comparative study of selective in vitro and in silico BACE1 inhibitory potential of glycyrrhizin together with its metabolites, 18α-and 18β-glycyrrhetinic acid, isolated from Hizikia fusiformis. Archives of Pharmacal Research, 41(4), pp.409-418.

Answer: this reference is talking about the activities of glycyrrhizin isolated from H. fusiformis, including its metabolites, 18α- and 18β-glycyrrhetinic acid against Alzheimer's disease (AD) via acetyl and butyrylcholinesterase and β-site amyloid precursor protein cleaving enzyme 1 (BACE1) inhibition.

Introduction:

Line 111, Here in the reference, there are other biological activity included. Is there any reason, authors focused only in this mentioned activity?

Answer: This sentence was revised.

Table 5, Compound 42, Fucosterol. The reference mentioned for this compound does not include any information regarding it.

Answer: This was revised and corrected

Similar happened in the table 6, The reference for the compound Dysidine (45, 65) does not match with content. The reference should be “Li, Y., Zhang, Y., Shen, X. and Guo, Y.W., 2009. A novel sesquiterpene quinone from Hainan sponge Dysidea villosa. Bioorganic & medicinal chemistry letters, 19(2), pp.390-392”.

Answer:  This was revised and corrected

Likewise, again in Table 6, the reference author cited does not have information for Dysidavarone A.

Answer: This was revised and corrected

Above mentioned are only few listed, however, there are plenty of such major mistakes. I would like to advice to go thoroughly through the manuscript and correct the mistakes.

The references mentioned in the Table 6 does not match with the text.Again, similar in case of Table 7 and 8 also.

Answer: All the tables references were revised and corrected

In figure 2, the name of the structure is incorrect. (2ʹ,3,5ʹ,6-pentabromo-3ʹ,4,4ʹ,5-tetrahydroxydiphenylmethane)

Answer: This was corrected

Minor comments:

There are a lot of grammatical errors in the text. Some of them along with the other errors as well have been mentioned in the following text.

Answer: Thank you for the reviewer advice. The manuscript was completely revised.

Abstract:

Line 39, “Structural-activity relationships (SARs) that could be drawn from the available data”, not “drown”.

Answer: Corrected.

Introduction:

Line 46, The reference provided for the statistical data should be latest one. For example; Reference 1- “The burden of mortality attributable to diabetes: Realistic estimates for the year 2000”. Similar condition applies to the reference 3, line 49.

Answer: The two references were updated

Line 57, “beneficial in improving the quality life of T2DM patients sound better than beneficial in improving the quality of life of T2DM patients.

Answer: Corrected.

Line 57-60, The sentence looks a bit unusual when read, especially after mechanism of action, such as insulin release stimulation……….”. Modify the sentence.

Answer: Revised.

Line 63, Rather than citing an article, it’s better to cite a book for reference 11.

Answer: A proper reference book was added. 

Line 95, “and being hydrophobic in nature” instead of “and being of hydrophobic nature”.

Answer: Revised.

Line 114, “Macroalgae species can reach 9.000 species?” There should be comma instead of fullstop.

Answer: Revised.

Line 132-134, This sentence should be cited with suitable reference. (The isolation and identification…………… possessing PTP1B inhibitory activity).

Answer: Done 

Line 142, “some of these compounds are discussed” instead of “is discussed”.

Answer: Revised.

In Table 1, Number 11 compound, there is no PTP1B inhibition activity mentioned.

Answer: The IC50value was added 

In Table 1, Compounds 14-16, The unit mentioned in the table is different with that of text at line 157-159.

Answer: Checked and revised.

Use same number of significant figures after every decimal place.

Answer: Checked and revised.

The symbol prime -“ ʹ ” is denoted differently in many place. Kindly, go through the manuscript and make it uniform throughout.

Answer: Revised

Table 2, Compound 23, make it uniform throughought the manuscript. IC50 2.66 µg/mL while others have IC50 = ….

Answer: Revised

Line 174, There is mentioned six polybromodiphenyl ether derivatives, but in table no any six polybrominated compounds seen.

Answer: Revised

Line 177, Its better to mention the significance of mentioning Huh-7 inhibition for compounds 24 and 25.

Answer: Revised and included a proper explanation of its pertinence.

Table 4. In this table, the IC50 value is along with the standard error of mean (2.64 ±0.04 µM), however, others table include only IC50 value. Make it uniform throughout the manuscript. And, there should be space between the … “±0.04”.

Answer: This aspect was revised throughout the manuscript.

What is the significance of the reference article 78? What are the points that are cited from this reference?

Answer: This reference was removed

In reference 83, regarding PTP1B, author mentioned as, “The potential of microalgae to block PTP1B activity was screened, but no apparent bioactivity was observed (data not shown)”. Here the meaning of the sentence is not clear.

Answer: This sentence was rewritten to be more clear

Line 274-277, The sentences and reference cited are quite confusing with each other. And the spelling of Symphyocladia latiuscula is incorrect in the manuscript (Line 274). Since, these are again explained in details in “Concerns in culture media”, here it can be modified mentioning only what is necessary over there.

Answer: Revised and corrected.

Line 339, “through using vanadium compounds” sounds quite strange. Better to write only “using”.

Answer: Revised

Line 342, Is it T1DM rats?

Answer: revised and corrected.

Line 374, Chromosome 20q13.1-q13.2 [102]…., it’s better to write chromosome 20 in the region q13.1-q13.2 or similar to that.

Answer: revised and corrected.

References:

The references are not aligned properly.

Use punctuations (comma, period and semicolon).

Check for the title of the study and make it in uniform pattern (either all words capital or small).

The author names are arranged randomly. Somewhere it is abbreviation and somewhere not. And also no use of proper punctuation and space bar. For example; 

Reference 58- M., T. N. M. a. A. T ???

Reference 60- Jung A. H., Y. N. Y, Woo M. which actually should be Jung H.A., Yoon N. Y., Woo, M. H and so on.

Reference 63- Abdul QA, C, R., Jung HA, Choi JS. , Heres it should be Choi R. J., and punctuation mark is missing in this reference. Follow same pattern within the manuscript.

Reference 79- the name J.S. is at last after the title of the article.

Reference 86- Kim, K. Y. N., K.A.;…. ????

(These are only some that has been enlisted. There are plenty of such mistakes in almost every references. I strongly recommend to go through the author’s guideline of the manuscript and follow the same pattern in each and every references)

Format for Molecules journal article:

Author 1, A.B.; Author 2, C.D. Title of the article. Abbreviated Journal Name YearVolume, page range. Available online: URL (accessed on Day Month Year).

(Similarly, see the journal website for others formatting).

The issue is mentioned in some places while missing in some. Follow journal guideline and do according. Make it uniform.

The page number range is missing. Check and correct it.

Somewhere there is use of journal abbreviations form while some have full name. Check and correct it.

The scientific name of plants and animals should be in italic. Check thoroughly.

The title is missing in reference 69.

I would suggest you to go along with the journal’s reference guideline. Most of the references are not in journal (Molecules) format. Check and correct.

Remarks: There are plenty of mistakes seen while referencing the manuscript. As a review article, these mistakes can lead to the decline of the true values of review paper. And also, the reference section, plenty of mistakes are seen. I would like to consider the above mentioned comments and make revisions accordingly.

Answer: all the highlighted aspects by the reviewer were carefully addressed.

Reviewer 2 Report

The authors presented a review on marine-derived bioactive molecules as upcoming anti-diabetic agents. The manuscript has a merit to be published in molecules. However, there are some suggestions which would improve the quality of the manuscript.

1. The authors should state in the text that there is no review which summarizes the same contents as this manuscript so far. If similar reviews existed, they should be cited.

2. Does the literature on marine-derived anti-diabetic agents compiled in this manuscript cover all the ones published so far on this topic? If not, the authors should state in the text what the reference was selected on the basis of what.

3. line 114; 9.0009000

4. line 21, Table 5; H-14 of sterols No. 36 and 37 should be deleted if it is to follow other compounds

5. line 221, Table 5; The structural formula of fecosterol (No. 42) should be modified following other sterols.

6. line 246, Table 6; The structural formula of sarsolilide B (No. 47) is incomplete. Clearly display the stereochemistry of C-10. Also, describe the hydroxy group in R of C-10.

7. line 330, Figure 2; The authors should summarize the compounds of Figure 2 in a table

8. The authors should carefully review the structural formulas of all the chemical moieties described in this manuscript.

Author Response

The authors presented a review on marine-derived bioactive molecules as upcoming anti-diabetic agents. The manuscript has a merit to be published in molecules. However, there are some suggestions which would improve the quality of the manuscript.

Answer: Thank you for the overall appreciation of our work.

1.The authors should state in the text that there is no review which summarizes the same contents as this manuscript so far. If similar reviews existed, they should be cited.

Answer: This information was added in introduction section (lines 99-100). 

2. Does the literature on marine-derived anti-diabetic agents compiled in this manuscript cover all the ones published so far on this topic? If not, the authors should state in the text what the reference was selected on the basis of what.

Answer: We only focused on marines having PTP1B inhibition activity

3. line 114; 9.000→9000

Answer: revised

4. line 21, Table 5; H-14 of sterols No. 36 and 37 should be deleted if it is to follow other compounds

Answer: Done

5. line 221, Table 5; The structural formula of fecosterol (No. 42) should be modified following other sterols.

Answer: Done

6. line 246, Table 6; The structural formula of sarsolilide B (No. 47) is incomplete. Clearly display the stereochemistry of C-10. Also, describe the hydroxy group in R of C-10.

Answer: Done 

7. line 330, Figure 2; The authors should summarize the compounds of Figure 2 in a table

Answer: This figure was removed and information included in a previous section according to the suggestion of reviewer 1.

8. The authors should carefully review the structural formulas of all the chemical moieties described in this manuscript.

Answer: Done

Round  2

Reviewer 1 Report

The references are not aligned properly.

Reference-It still needs corrections.

Use punctuations (comma, period and semicolon).

Check for the title of the study and make it in uniform pattern (either all words capital or small).

The author names are arranged randomly. Somewhere it is abbreviation and somewhere not. And also no use of proper punctuation and space bar. For example;

Reference 58- M., T. N. M. a. A. T ???

Reference 60- Jung A. H., Y. N. Y, Woo M. which actually should be Jung H.A., Yoon N. Y., Woo, M. H and so on.

Reference 63- Abdul QA, C, R., Jung HA, Choi JS. , Heres it should be Choi R. J., and punctuation mark is missing in this reference. Follow same pattern within the manuscript.

Reference 79- the name J.S. is at last after the title of the article.

Reference 86- Kim, K. Y. N., K.A.;…. ????

(These are only some that has been enlisted. There are plenty of such mistakes in almost every references. I strongly recommend to go through the author’s guideline of the manuscript and follow the same pattern in each and every references)

Format for Molecules journal article:

Author 1, A.B.; Author 2, C.D. Title of the article. Abbreviated Journal Name Year, Volume, page range. Available online: URL (accessed on Day Month Year).

(Similarly, see the journal website for others formatting).

The issue is mentioned in some places while missing in some. Follow journal guideline and do according. Make it uniform.

The page number range is missing. Check and correct it.

Somewhere there is use of journal abbreviations form while some have full name. Check and correct it.

The scientific name of plants and animals should be in italic. Check thoroughly.

The title is missing in reference 69.

I would suggest you to go along with the journal’s reference guideline. Most of the references are not in journal (Molecules) format. Check and correct.

Remarks: There are plenty of mistakes seen while referencing the manuscript. As a review article, these mistakes can lead to the decline of the true values of review paper. And also, the reference section, plenty of mistakes are seen. I would like to consider the above mentioned comments and make revisions accordingly.